# Analysis of the Informal Reasoning Modes of Preservice Primary Teachers When Arguing about a Socio-Scientific Issue on Nuclear Power during a Role Play

**Isabel María Cruz-Lorite** [1,*] **, Daniel Cebrián-Robles** [1] **, María del Carmen Acebal-Expósito** [1] **and Maria Evagorou** [2]

1.  Science Education Department, University of Málaga, 29016 Málaga, Spain
2.  Department of Education, University of Nicosia, 2417 Nicosia, Cyprus
*   Correspondence: imclorite@uma.es

**Abstract:** The use of nuclear power is a socio-scientific issue that is controversial in many areas. Concerns of nuclear power touch on health effects, environmental impacts, employment concerns and energy supply; arguments both for and against it are easily generated. This paper examines the specific aspects addressed by preservice primary teachers in their arguments during their participation in a roleplaying activity on the SSI of nuclear power stations closures in Spain. This was done in order to better understand informal reasoning modes, as well as the possible effect of the roles defended and the design of the staging of the role play. To this end, the transcripts of four role plays were analysed. The data analysis was carried out by open coding, extracting different categories of analysis that were classified into three different informal reasoning modes: environmental, financial and social. The results showed that participants used more environmental-oriented arguments than financial and social ones. Differences in informal reasoning modes were found between some roles and between the two parts of the staging. Some educational implications of these results are discussed, such as providing more information to the participants before the roleplaying activity and emphasising the scaffolding of the social aspects when designing the role play.

**Keywords:** socio-scientific issues; informal reasoning modes; roleplaying game; preservice primary teachers; nuclear power

## 1. Introduction

This paper aims to contribute to the knowledge of the informal reasoning modes used by preservice primary teachers (PPTs) when addressing a socio-scientific issue (SSI) related to the use of nuclear power during their participation in a roleplaying activity. In particular, the specific aspects addressed by PPTs in their arguments during the staging of a roleplaying activity are analysed to better understand the informal reasoning modes and the possible impact on them of the roles defended and of some aspects of the roleplay design.

This work provides some novel contributions to this field. Firstly, a more detailed analysis about the aspects addressed the PPTs' arguments is carried out, which can help to better define the informal reasoning modes to be considered in the analysis of arguments on the SSI of the use of nuclear power. Most of the studies analyse the informal reasoning modes from broad scope, such as ecological, economic, social and scientific-technological, but few studies have carried out a more detailed analysis of the specific aspects addressed in each of them (e.g., [1]). This can be one of the difficulties for achieving greater consensus on the establishment of the informal reasoning modes. In this paper, we understand specific aspects as the topics/categories that emerge from the SSI argumentation, for example, security, employment, politics, pollution, etc. These aspects are associated with informal reasoning modes to facilitate their analysis and interpretation. Secondly, the present work makes contributions to the existing knowledge about PPTs' oral informal reasoning modes.

Most of these and other studies on how PPTs deal with argumentation about SSI analyse written productions [1–4], with few studies analysing PPTs' oral skills (e.g., [5]).

A review of the literature in relation to the most relevant aspects involved in this research is presented below.

### 1.1. The Use of Nuclear Power as a Socio-Scientific Issue

According to Patronis et al. [6], societies that claim to be based on advanced technology present the democratic problem that only a limited group of people seem to be able to handle the complexity associated with scientific-technological knowledge. To manage this complexity, it is important that citizens develop their argumentative skills to be able to participate in public debates and in the decision-making processes to solve complex problems [7]. This raises the need to deal with such complex problems in the classrooms, one of the most widely used approaches in science education being the treatment of SSI, which has become an important component of this field [8].

SSI are ill-structured problems that involve moral, ethical and financial aspects, and lack clear-cut solutions, which usually emerge from the nexus of science and society [9]. The treatment of SSI in the classrooms has shown positive results in the development of scientific content understanding [10–13], learning about the nature of science [14] and more interest and motivation to learn when science is presented in socially relevant contexts [15–18]. Hogan [19] also points out that discussing controversial environmental issues is an engaging way to encourage students' thinking.

The use of nuclear power is considered an SSI, which raises public debates [20] and seems to be relevant to decisions students may contribute as responsible citizens [21]. Political, social, economic, scientific-technological and environmental aspects, among others, converge in an increasingly relevant issue, which requires citizens to take a stance, given that it is, together with renewable energies, one of the main alternatives to fossil energy sources [22]. The controversy about the use of nuclear power can be seen in the sometimes-conflicting positions taken by some countries in recent decades. In Europe, some countries (exemplified by France and Finland) show pronuclear positions while others (e.g., Spain and Germany) show antinuclear positions [23]. Public opinion in Europe also appears to be evenly split on the issue of nuclear power [24]. However, in 2022, the European Parliament approved labelling some nuclear power projects 'green', a decision that compromises the position of countries such as Spain, which already has a plan to decommission all its nuclear power stations [25].

### 1.2. Roleplay, Argumentation and SSI

Given their interdisciplinary nature, effectively addressing SSI, such as the use of nuclear power, in science classrooms requires teachers to encourage students to examine and express different points of view on the problem [26]. This makes roleplaying activities particularly suitable strategies for dealing with these problems, given that its main objective is the representation of real situations, with each participant defending a role [27], comprising the appropriate characteristics for the practice of argumentative skills on SSI. Several studies report benefits in the use of roleplaying activities, in science teaching (e.g., [28–35]). The literature also shows that roleplaying helps to improve the processes of argumentation and refutation of opposing views [36], and the argumentative skills are considered fundamental for the 21st-century citizens of democratic societies [37] to face contemporary SSI. Argumentation about SSI should have a central role in science teaching because it reflects the endeavour of negotiating and constructing knowledge that takes place in science [38]. Therefore, the use of argumentation and debate activities, such as roleplaying, is a useful means of engaging thought and reasoning processes [39].

Argumentation has been addressed in the science teaching classrooms from a variety of perspectives and educational stages. However, few studies have focused on the assessment or development of preservice teachers' understanding of argumentation [40]. The promotion of argumentative activities with preservice teachers is an important step

to introduce argumentation in basic education [41], and Chan and Erduran [42] point out that "it would not be prudent to rely on teachers to promote argumentation in classrooms if they themselves have no experience in engaging in scientific argumentation" (p. 4). Given that the PPTs' experience from the student's point of view is an essential part of their training, roleplaying activities are ideal didactic strategies for introducing PPTs to argumentation about SSI and a way to reinforce their future teaching practices. However, some studies have pointed out the low-quality argumentation skills of preservice science teachers (e.g., [1,20]), and others report that one of the challenges faced by teachers in SSI teaching is the lack of knowledge about SSI [43]. In the case of nuclear power, Borgerding and Dagistan [44] carried out a study with one inservice and 12 preservice secondary teachers, reporting that "many participants identified nuclear power as controversial and one they felt ill-prepared to teach" (p. 301). Therefore, assessing both the PPTs' argumentative skills and their knowledge about SSI is key for developing training programs that help to reinforce their future teaching practice.

### 1.3. Informal Reasoning about the SSI of Nuclear Power

SSI differ from other scientific questions, with SSI usually being related to informal reasoning and other scientific questions to formal reasoning [45]. While formal reasoning is used in formal contexts, where problems are well defined, premises are always explicit and clear, and there are definite conclusions or solutions, informal reasoning is applied outside formal contexts, where problems are not well defined, premises may not be explicitly stated and, consequently, derived conclusions may not be easily delimited [46]. Therefore, arguing about SSI implies the use of informal reasoning processes.

In the case of the use of nuclear power, a review of the literature on preservice teachers' informal reasoning shows some differences for analysing the domain to which the arguments are oriented (e.g., ecological, economic, social, etc.). Firstly, while Demircioğlu and Uçar [2], Ozturk and Yilmaz-Tuzun [1] and Saglam and Eroglu [4] use the term 'reasoning mode', Evren and Aycan [3] and Crujeiras et al. [5] use the terms 'informal reasoning type' and 'type of knowledge', respectively. The term reasoning mode is also used by other authors in studies with high school students (e.g., [47–49]). Following the majority consensus, in this work we also use the term 'reasoning mode'.

Table 1 shows the different reasoning modes established in works on preservice teachers' argumentation about SSI related with nuclear power. Although there are differences in the number of reasoning modes considered, all of them coincide in four reasoning modes: ecological, economic, social and scientific-technological.

**Table 1.** Works on preservice teachers' informal reasoning about SSI related to nuclear power and modes established.

| | Demircioğlu and Uçar [2] | Ozturk and Yilmaz-Tuzun [1] | Evren and Aycan [3] | Crujeiras et al. [5] | Saglam and Eroglu [4] |
|---|---|---|---|---|---|
| Ecological | X | X | X | X | X |
| Economic | X | X | X | X | X |
| Social | X | X | X | X | X |
| Scientific-technological | X | X | X | X | X |
| Types of risks | | X | | | |
| Politics | | X | | | |
| Moral and ethics | | | | X | |

Note: In all the cited works, the sample consisted entirely, or a large percentage, of PPTs. The only exception is in the work of Evren and Aycan [3], which do not specify the type of preservice science teachers participating, only their age range (19–21), which is the same as the age range of the sample in this study.

These studies also differ in their results. On the one hand, some works point out that the ecological reasoning mode plays a relevant role in the preservice teachers' arguments about nuclear power. Demircioğlu and Uçar [2] carried out a study about the reasoning mode used by 38 PPTs about the construction of a nuclear power station. PPTs took a pre-test and a post-test before and after reading an article about the construction of a nuclear power station. The results showed that in both the pre-test and the post-test, the

most used type of reasoning was ecological, while social reasoning was the least used. In the same line, Ozturk and Yilmaz-Tuzun [1] analysed the argumentation skills of 674 PPTs about the building of a nuclear power station. The authors established six categories for the reasoning modes, which in proportion were expressed by the PPTs in the following order: ecology-oriented, economic-oriented, types of risk, social-oriented, political-oriented and science/technology-oriented arguments. Evren and Aycan [3] analysed the type of informal reasoning of 51 third-year students attending the department of science teaching before and after the application of SSI-based instructional activities about the establishment of nuclear power stations. Before the activity, students most used ecological type of informal reasoning, followed by scientific-technologic, social and, finally, economic. However, after the activities the order was social, ecological, economic and scientific-technologic.

On the other hand, other works seem to find different predominant PPTs' informal reasoning modes. Crujeiras et al. [5] analysed the oral argumentative skills of 112 preservice teachers (87 PPTs and 25 preservice preschool teachers) during their participation in a roleplaying activity about the installation of a nuclear cemetery. In this case, the arguments were analysed from the perspective of the type of knowledge used in their construction, with the most common type of knowledge being social, followed by scientific-technological, environmental, moral and ethical, and finally, economic. Saglam and Eroglu [4] carried out a study with undergraduate students from the departments of science teacher education (31), primary school teacher education (41) and social science teacher education (28). This study qualitatively analysed the informal reasoning mode used in students' written responses to open-ended questions on the building of nuclear power stations. Their results showed that the social informal reasoning mode was the most used, followed by the ecological and economical ones, while scientific/technological reasoning was the least used.

As can be seen, the literature on informal reasoning about nuclear power used by preservice teachers and, specifically, by PPTs, shows different results, both in terms of the number of informal reasoning modes and in terms of their results, which indicates the need for further studies in this field. For all of the above, the aim of this paper is to study what specific aspects the PPTs address when participating in a role-play activity about the use of nuclear power in order to better understand the informal reasoning modes used by them. The research questions established are as follows:

1.  What aspects of SSI on the closure of nuclear power stations in Spain and informal reasoning modes do PPTs use in their arguments during a staging of a roleplay, and how often do they do so?
2.  Is there a relationship between the roles and the frequency in which the informal reasoning modes are used?
3.  Are there differences in the frequency of the informal reasoning modes used between the two parts of the staging of the roleplay?

## 2. Methodology

This study is non-experimental with a multidimensional idiographic design within a mixed approach [50]. Therefore, a qualitative analysis was carried out to extract data from the roleplaying transcripts, followed by a quantitative analysis to summarise the results in order to allow comparisons and the application of statistical tests.

The students participating in this study, as indicated, were of legal age (over 18 years old in Spain) and informed consent was obtained from all of them. The participants were informed that their data would be treated exclusively for research purposes, and that it would be impossible to identify them in the publications derived from the study, following the recommendations of BERA's Ethical Guidelines for Educational Research [51].

### 2.1. Participants

The experiences were carried out with 125 PPTs (class group A, n = 69; and B, n = 56) enrolled in a Teaching Science module during the 2018–2019 academic year. Most of the participants were 20 years old. Each one of these class groups were divided into two groups.

Therefore, a total of four roleplays were carried out (hereinafter referred to as A1, A2, B1 and B2). The PPTs were in year three of the Bachelor's in Primary Education, a four-year degree offered by the University of [BLINDED FOR REVIEW]. Some PPTs did not attend the staging sessions of the roleplays. Therefore, the number of PPTs who participated in the stagings was 104.

### 2.2. Teaching Context

This study has mainly focused on the learning-to-argue approach [52], which focuses on the 'acquisition of argumentation skills, such as constructing valid arguments, selecting appropriate evidence to justify those arguments, identifying and constructing counterarguments, and responding appropriately to those counterarguments' [53] (p. 3). For this reason, before their participation in the roleplaying activity the PPTs participated in a sequence of tasks to learn to teach and assess argumentation developed by Cebrián-Robles et al. [54]. The roleplaying activity consisted of three parts: (i) introduction to the roleplaying activity and assigning roles to students; (ii) preparation of roles by students; (iii) the roleplay staging. The staging had two parts: the first was a presentation round, in which each role had three minutes to present its position and initial arguments, and in the second part there was an open debate. The roleplay design used is the result of design-based research developed in previous research [55], where the full description of the roleplaying activity can be found.

The roleplaying activity was about the agreement reached by the Spanish government and the country's major electricity companies to gradually phase out nuclear power production over the period 2025–2035 [25]. The PPTs were organised in small groups between three and four students, each of them had to defend one role. In each small group, one of the PPTs was the spokesperson while the rest were the advisory team, with different tasks during the staging of the roleplaying activity. To prepare the roles, they received cards with the descriptions of different roles. This descriptions reflected three different positions about the pact which come from a position most strongly in favour of nuclear power to a position most strongly against: (1) no agreement with the pact, nuclear power production should continue indefinitely (position A); (2) agreement with the pact, nuclear power production should to phase out over the period 2025–2035 (position B); or (3) no agreement with the pact, nuclear power production should cease immediately, in 2020 (position C). For the staging of the roleplaying activity, the roles were organised into two groups: the group in favour of the use of nuclear power (roles in position A) and the group against the use of nuclear power (roles in positions B and C). The roles, their acronyms, the different positions about the pact and the groups for the staging can be seen in Figure 1.

Given that some PPTs did not attend the staging sessions, the role of the politician from the opposition was only represented in one of the roleplaying activities, whereas the role of ecologist was not represented in any of them.

### 2.3. Data Collection and Analysis

Data were collected by video recording the sessions in which the roleplaying activities were carried out. All the video recordings were transcribed for further analysis. The text of the transcripts, consisting of 29,797 words, was organised providing the time in which an intervention starts in the video recording and the identification of the person who was talking. Therefore, each piece of text spoken by a single person was considered an utterance. The utterances have a variable length and they could contain one or more arguments. Furthermore, there were three interventions that it was not possible to identify who had made them, due to problems with the recording. These interventions were not considered in the analysis. Similarly, only five interventions by the presenters were recorded relating to the SSI, as their task was mainly to manage and lead the staging. Therefore, this role was also not considered in the analysis of the results. Finally, some of the members of the advisory teams participated orally in the staging, although these interventions were

not frequent. Both the interventions of the spokespersons and those of the members of the advisory teams were analysed together as interventions of a certain role.

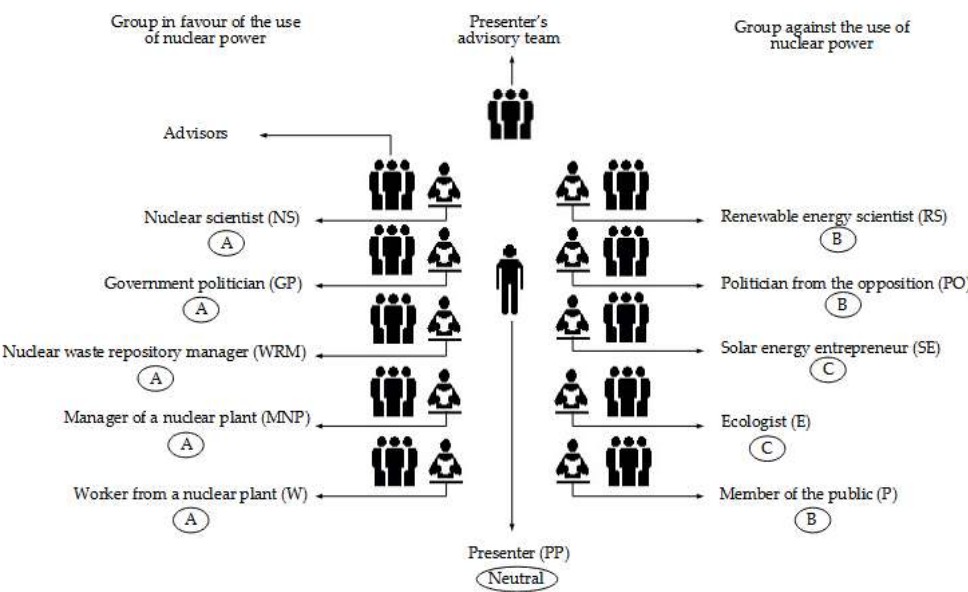

**Figure 1.** Roles, acronyms, the different positions held and the groups formed for the staging.

### 2.4. Category System

Although there is agreement on some of the informal reasoning modes that PPTs use when arguing about nuclear power-related SSI, the literature shows a diversity of modes. It was therefore decided in this paper to carry out a process of open coding of the specific aspects addressed in the arguments in order to subsequently form the modes. As a result of this analysis, 29 categories were obtained and grouped into three informal reasoning modes: environmental, financial and social. Although the PPTs included scientific-technological information or knowledge in their interventions, a scientific-technological informal reasoning mode was not established, as scientific-technological knowledge was not expressed in the roleplaying activity in isolation, but as part of the arguments that the PPTs used to address environmental, economic or social modes of reasoning about the SSI addressed.

### 2.5. Analysis Procedure

The roleplaying activities were carried out in Spanish. For the initial analysis, one of them was translated to English, due to one of the researchers involved in this study does not speak Spanish. The rest of the roleplaying activities were analysed in the original language, as it was considered more appropriate. Only a few fragments were translated in order to discuss those cases where there was disagreement among researchers in order to reach a consensus.

For data analysis, first, two of the authors conducted several rounds of analysis to create the category system on the aspects included in the arguments. These rounds consisted of two rounds of analysis of the roleplay B1 and one round of analysis of the roleplay B2. The categories found were not mutually exclusive. A first version of the category system consisting of 18 categories organised into three informal reasoning modes was obtained. For the calculation of inter-rater reliability, Wimmer and Dominick [56] recommend that a sub-sample of between 10% and 25% of the data be analysed by independent coders. For this reason, 28 utterances from the roleplay A1 (25% of the text transcribed for this roleplay) were selected. This percentage of text includes utterances of the end of the presentation round and utterances of the beginning of the debate. Three of the authors analysed these utterances and then they were compared. The agreement was calculated by percentages and Kappa coefficients for the 18 categories of the first version of the analysis framework. All

the agreement percentages were higher than 75%. Kappa coefficient was calculated for all the variables studied, obtaining values close to zero in all cases. The cause for these results is known: the low prevalence of the categories makes that, despite the high agreement percentage found, the values for Kappa coefficient were low [57]. The four roleplaying activities were analysed with the final analysis framework and the disagreements were resolved through discussion among all authors. The qualitative analysis was carried out with the qualitative analysis software Atlas.ti v.9.

The results were quantitatively analysed calculating the absolute frequencies and the percentages of the different aspects studied. They were organised in contingency tables for comparisons and for conducting statistical tests with the software R-commander v.4.2. The chi-square test was used to analyse the possible existence of statistically significant differences and the significance level was set at 0.05 for all tests. In the case of the existence of statistically significant differences the effect size (ES) was calculated for the magnitude of differences, using the Cramer's V (V-ES) method [58]. The Cramer's V is interpreted as small, moderate or large depending on the degrees of freedom of the rows or columns (whichever is less) of the contingency table. The ranges used for interpretation can be consulted in López-Martín and Ardura-Martínez [59].

## 3. Results

### 3.1. SSI Aspects on Nuclear Power and Informal Reasoning Modes

After analysing all the roleplaying activities, 29 categories of SSI aspects were extracted and grouped in three informal reasoning modes: environmental, financial and social. Examples of quotes included in each one of the categories can be consulted in Appendix A. The results of the analysis, the categories and their descriptions can be seen in Table 2. Environmental-oriented arguments are the most used (362 arguments), followed by the financial ones (174), and social-oriented arguments are discussed to a much lesser extent (102). The categories with more frequencies are pollution, safety, employment, energy supply, production costs, waste and waste management, health, resources depletion and continuous production, in that order.

### 3.2. Roles and Informal Reasoning Modes

There is a similar behaviour in all roles in terms of the informal reasoning mode used in their contributions (Figure 2). In general, most of their arguments were environmental-oriented, followed by the financial-oriented and social-oriented arguments. Five roles (P, E, PO, RS and NPM) follow the same patterns shown in the general results (environmental > financial > social). However, the roles SE, GP, WRM, W and NS show different patterns: (1) SE and GP have more environmental-oriented arguments, then social-oriented and finally financial-oriented arguments; (2) WRM and W have more a similar number of environmental-oriented and financial-oriented arguments than the other roles; and (3) NS has the same number of financial-oriented and social-oriented arguments. The role with the most balanced use of informal reasoning modes is the GP.

To study the possible statistical differences between the ten roles (the role of PP is not considered) and the three informal reasoning modes, the chi-square test was applied without considering the category 'Unknown aspect' ($10 \times 3$ contingency table). There were statistically significant differences with a small effect size ($\chi^2 = 47.088$; $p = 0.000$; V-ES = 0.193). Statistically significant differences found in pairwise comparisons ($3 \times 2$ contingency tables) can be seen in Table 3. The role that shows a higher number of statistically significant differences is GP (differences with 8 roles), the second ones are SE and NS (7), and the rest have two or less. All the statistically significant differences were with a small effect size.

**Table 2.** Number of arguments classified in each category, their descriptions and the informal reasoning mode to which they were associated.

| Informal Reasoning Mode | Category | Related to . . . | Total |
|---|---|---|---|
| Environmental | Pollution | How a certain energy source pollutes the environment, both by chemical or physical means. | 150 |
| | Safety | How much safe a certain energy source is | 81 |
| | Waste and waste management | The waste that a certain energy source emits and how it is managed | 44 |
| | Health | Human health aspects about using a certain kind of energy source | 29 |
| | Resources depletion | The reserves of natural resources or the consumption pace | 28 |
| | Climate change | The effects of a given energy source on the climate change | 12 |
| | Environmental | Environmental aspects in general | 12 |
| | Resources availability | The availability of natural resources in certain places or conditions | 6 |
| Total | | | 362 |
| Financial | Employment | What suppose a certain energy source in terms of employment | 62 |
| | Production costs | How much it costs to produce/distribute a certain type of energy (it includes the costs of decommissioning nuclear power stations or nuclear cemeteries, for example) | 46 |
| | Continuous production | How feasible is to maintain a continuous energy production with certain energy source | 23 |
| | Financial | Financial aspects in general | 9 |
| | Dependence on other countries | The dependence on other countries to supply the energy demand | 8 |
| | Saving money | How much money we could save using a certain energy source | 7 |
| | Earn money | The incomes for towns, cities, countries, etc., of the use of a certain energy source | 6 |
| | Economic interests/corruption | Economic interests and corruption associated with energy production | 6 |
| | Price stability | The fluctuation of the prices | 5 |
| | Viability | Practical/economic viability of a certain energy source | 2 |
| Total | | | 174 |
| Social | Energy supply | How to supply the energy demand of the society and/or about why we need to use a certain energy source (or other things, for example nuclear cemetery) to supply the society | 61 |
| | Politics | The role of politics in the use of a certain energy source (e.g., taxes for the public) | 14 |
| | Technological development | The development/benefits/detriments that technology in relation to energy production produces in society | 7 |
| | Nuclear weapons and terrorism | The use of nuclear energy for war or terrorist purposes | 6 |
| | Resettle | The benefits of nuclear power for depopulated areas | 5 |
| | Awareness/perception on nuclear power | People's awareness/perception on nuclear power and/or the need to change their awareness | 3 |
| | Reduce consumption | The need for society to reduce its energy consumption | 2 |
| | Media | The media treatment of different energy sources | 2 |
| | Other uses | Other uses of nuclear energy (e.g., medicine, food conservation, archaeology or C-14 test) | 2 |
| | Social | Social aspects in general | 0 |
| Total | | | 102 |
| | Unknown aspect | It is not possible to know what the aspect is exactly that they talk about | 18 |

For studying the possible statistically significant differences, the data were grouped according to the two role groups, in favour and against, organised for the staging (Figure 3). There is virtually no difference between roles in favour and against in using the three informal reasoning modes and the chi-square test indicated that there were not statistically significant differences ($\chi^2 = 1.175$; $p = 0.556$).

Figure 3 appears to show a relationship between informal reasoning modes and the in favour and against groups. This indicates that both groups of roles argued using roughly the same number of arguments from the three informal reasoning modes, following the order: environmental > financial > social. The differences that are noticeable (not statistically significant) are that the group against makes more environmental-oriented arguments, while the group in favour makes more financial-oriented and social-oriented arguments. It seems that the pro-nuclear roles have relied on these types of arguments more than the anti-nuclear roles.

*3.3. Informal Reasoning Modes in the Two Parts of the Staging*

In the presentation round, where there were no interruptions and usually no counterarguments, there was a better balance between environmental-oriented and financial-oriented arguments (Figure 4). It was during the debate that more social-oriented arguments appear. Perhaps this indicates a shift in argument strategy, seeing that it might be more effective to launch social-oriented arguments.

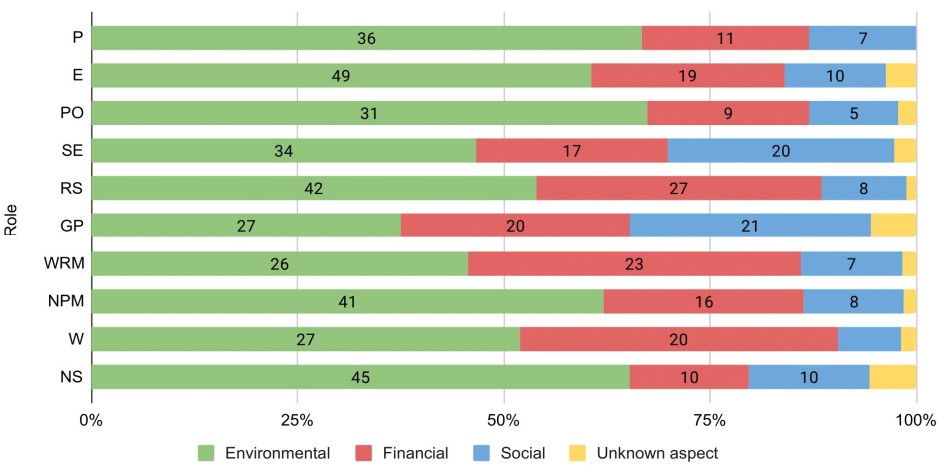

**Figure 2.** Results of the informal reasoning modes used by each role. The data labels shown in the bars correspond to the results in absolute terms. Absolute frequency labels less than 5 are not included. Note: abbreviations: member of the public (P); ecologist (E); politician from the opposition (PO); solar energy entrepreneur (SE); renewable energy scientist (RS); government politician (GP); nuclear waste repository manager (WRM); manager of a nuclear plant (NPM); worker from a nuclear plant (W); and nuclear scientist (NS).

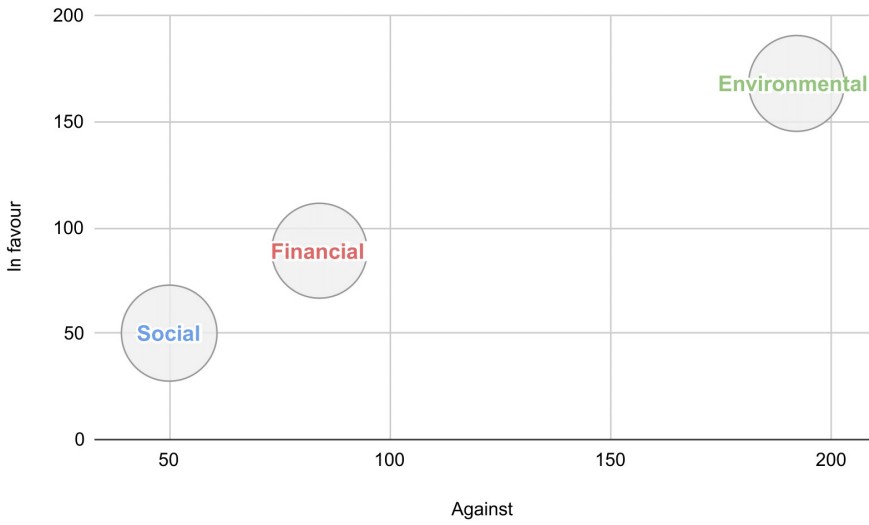

**Figure 3.** Frequencies of the informal reasoning mode used depending on the position of the roles.

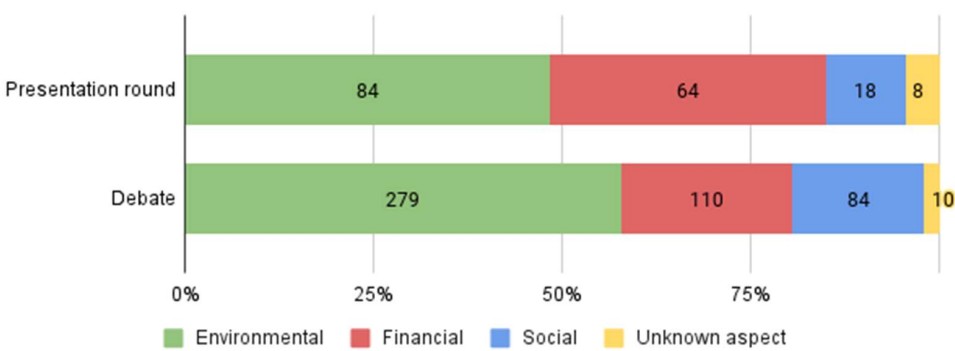

**Figure 4.** Absolute frequencies and percentages of the informal reasoning modes depending on the part of the staging.

**Table 3.** Statistically significant differences and effect sizes found for pairwise comparisons between the roles.

| | | P | E | PO | SE | RS | GP | WRM | NPM | W | NS |
|---|---|---|---|---|---|---|---|---|---|---|---|
| P | $\chi^2$ | | | | | | 9.416 | | | | |
| | *p*-value | | | | | | 0.009 | | | | |
| | V-ES | | | | | | 0.278 | | | | |
| E | $\chi^2$ | | | | | | 9.658 | | | | |
| | *p*-value | | | | | | 0.008 | | | | |
| | V-ES | | | | | | 0.257 | | | | |
| PO | $\chi^2$ | | | | 6.078 | | 10.028 | | | | |
| | *p*-value | | | | 0.048 | | 0.007 | | | | |
| | V-ES | | | | 0.229 | | 0.298 | | | | |
| SE | $\chi^2$ | | | | | 8.028 | | 6.546 | | 8.667 | 6.428 |
| | *p*-value | | | | | 0.018 | | 0.038 | | 0.013 | 0.040 |
| | V-ES | | | | | 0.233 | | 0.227 | | 0.267 | 0.217 |
| RS | $\chi^2$ | | | | | | 9.609 | | | | 7.174 |
| | *p*-value | | | | | | 0.008 | | | | 0.028 |
| | V-ES | | | | | | 0.257 | | | | 0.225 |
| GP | $\chi^2$ | | | | | | | 6.124 | 9.091 | 9.322 | 11.675 |
| | *p*-value | | | | | | | 0.047 | 0.011 | 0.010 | 0.003 |
| | V-ES | | | | | | | 0.222 | 0.261 | 0.280 | 0.296 |
| WRM | $\chi^2$ | | | | | | | | | | 10.122 |
| | *p*-value | | | | | | | | | | 0.006 |
| | V-ES | | | | | | | | | | 0.289 |
| NPM | $\chi^2$ | | | | | | | | | | |
| | *p*-value | | | | | | | | | | |
| | V-ES | | | | | | | | | | |
| W | $\chi^2$ | | | | | | | | | | 8.844 |
| | *p*-value | | | | | | | | | | 0.012 |
| | V-ES | | | | | | | | | | 0.276 |
| NS | $\chi^2$ | | | | | | | | | | |
| | *p*-value | | | | | | | | | | |
| | V-ES | | | | | | | | | | |

Note: abbreviations: member of the public (P); ecologist (E); politician from the opposition (PO); solar energy entrepreneur (SE); renewable energy scientist (RS); government politician (GP); nuclear waste repository manager (WRM); manager of a nuclear plant (NPM); worker from a nuclear plant (W); nuclear scientist (NS); chi-square ($\chi^2$); Cramer's V effect size (V-ES).

Statistically significant differences with a small effect size were found between the two parts of the staging ($\chi^2 = 15.763$; $p = 0.000$; V-ES = 0.157). In both the presentation round and the debate, there are more environmental-oriented arguments than financial-oriented and social-oriented arguments. However, while the number of financial-oriented arguments increases 1.72 times in the debate compared to the presentation round, the environmental-oriented and social-oriented arguments increase 3.32 and 4.67 times, respectively.

## 4. Discussion

The findings show that there is a great imbalance between the informal reasoning modes used in the staging, with a predominance of environmental-oriented arguments, followed by financial-oriented arguments, the social-oriented arguments being the least addressed. These results are consistent with those reported by Demircioğlu and Uçar [2], who found that the ecological-oriented arguments were the most generated by PPTs, while the least were the social-oriented ones. They also fit with the results obtained by Ozturk and Yilmaz-Tuzun [1], who found that the arguments built by the PPTs were mostly ecology-oriented, followed by economic-oriented. However, in this case, social-oriented arguments were not the least addressed. Evren and Aycan [3] also found that PPTs most used the ecological type of informal reasoning before the application of SSI-based instructional activities, although after the activity PPTs most used the social type of informal reasoning. However, these results differ from those obtained by Crujeiras et al. [5] and Saglam and Eroglu [4], who mainly found social-based arguments. This may be due to the contextual differences between these studies and ours. Thus, Crujeiras et al. [5] addressed the problem of the installation of a nuclear cemetery, in which the social-based arguments seem to

acquire more prominence. In the case of the study by Saglam and Eroglu [4], although the problem is the same, the differences may be due to the format in which the argumentation takes place, in writing and individually, whereas in our study it takes place in the context of an oral debate. Although a scientific-technological informal reasoning mode has not been analysed in this study, there were explicit mentions of science and technology in some arguments (see Appendix A).

No statistically significant differences were found in terms of the informal reasoning modes used by the different roles. However, the fact that the roles of ecologist and politicians make the largest number of environmental-oriented and social-oriented arguments, respectively, makes sense given the role profiles. Furthermore, roles against nuclear power, except for one, maintain a marked difference in the number of contributions to the informal reasoning modes, following the order environmental > financial > social, while in the roles in favour of nuclear power, except for one, these differences are lower or even change the order of the number of contributions to them. This could indicate that roles against the use of nuclear power focus their argumentation on environmental issues, while roles in favour of the use of nuclear power need to provide more arguments on financial and social issues. However, no statistically significant differences were found in the number of contributions to each informal reasoning mode depending on the position (in favour or against) of the roles.

Finally, although in both parts of the staging, the presentation round and the debate, there are mainly environmental-oriented arguments, followed by financial-oriented and social-oriented arguments, environmental and social ones increase more than the financial ones in the debate with respect to the presentation round. We interpret these results in the sense that the debate, the purpose of which is to refute the arguments of other roles, PPTs, taking into account the high degree of concern of young people for environmental issues [60], may consider the environmental-oriented and social-oriented arguments to be more persuasive for them than financial-oriented arguments. Saglam and Eroglu [4] showed that the preservice teachers that participated in their study relied on social-oriented arguments to develop their supportive arguments, whereas when refuting they relied on ecological-oriented arguments. In our study, the presentation round of the staging is mainly for presenting the initial arguments, being the debate the part of the staging in which the PPTs have to elaborate counterarguments and rebuttals. Therefore, in this sense, our results only partially coincide with those reported by Saglam and Eroglu [4]. Moreover, our results are in part contradictory to those reported by these researchers, since if interpreted in incremental terms, social arguments experienced the highest increase in the debate part (4.67 times more than in the presentation round).

## 5. Conclusions and Implications

The main conclusions of this study are as follows:

1. Regarding research question 1 (i.e., what aspects of SSI on the closure of nuclear power stations in Spain and informal reasoning modes do PPTs use in their arguments during a staging of a roleplaying activity, and how often do they do so?), the findings of this study show that PPTs arguments are focused on environmental aspects and, to a lesser extent, on financial aspects; the least addressed are social aspects.

2. Regarding research question 2 (i.e., is there a relationship between the roles and the frequency in which the informal reasoning modes are used?), there are statistically significant differences that allow us to establish the hypothesis of the existence of differences between the roles in terms of the informal reasoning mode dealt with during the staging. Practically all the roles against the use of nuclear power follow a marked structure in the proportions of the informal reasoning mode used during the staging (environmental > financial > social), while practically all the roles in favour of these differences in the proportions are reduced or changed. It seems that the roles in favour of the use of nuclear power must strengthen their positions by increasing the number of financial-oriented and social-oriented arguments compared to those shown by the roles against. However, there are no statistically significant differences

that would allow us to say that there are two distinct groups. Furthermore, the roles GP, SE and NS seem to deviate from the argumentation patterns followed by the other roles, considering the significant differences found.

3. Regarding research question 3 (i.e., are there differences in the frequency of the informal reasoning modes used between the two parts of the staging of the roleplaying activity?), there are statistically significant differences between the informal reasoning mode used during the two parts of the staging. Social-oriented arguments are the ones that increase the most, in proportion, in the debate compared to the presentation round, followed by environmental-oriented and finally financial-oriented arguments.

In the opinion of the authors, the educational implications of the findings presented in this work go in the direction of providing more information to the PPTs before the staging. When designing the roleplaying activity, it seems necessary to put more emphasis on scaffolding the social aspects. With respect to this implication, Sadler and Zeidler [45] point out that, based on experiences with their students, teachers can frame SSI in ways that encourage different modes of informal reasoning. On the one hand, it is important to know the predominant informal reasoning mode when selecting an SSI, as it can condition the roles and the roleplaying activity to argue mostly on specific aspects, moving away from other equally relevant areas when dealing with that SSI. Big issues such as energy supply for a growing world population, people's access to energy self-sufficiency through renewable energies or political decisions about the use of nuclear power are overlooked in the staging when compared to the attention paid to environmental and financial aspects. To help balance the debate in terms of topic areas, PPTs could be offered as additional information before preparing their roles, so that their search for information is not only directed towards the aspects most present in the media (which tend to be environmental and, at a lesser extent, financial), but also towards others with equal relevance to understanding the societal scope of this SSI. This information could come in the form of a brief summary of the controversies surrounding the use of nuclear power and/or the provision of further information on the role description cards, trying to ensure that the amount and nature of the information contained in them does not hinder an open and free-flowing debate [61]. In addition, the open debate (second part of the staging) has been found to be conducive to the contribution of arguments on social aspects.

The possible implications of this study, given the nature of the problem addressed, go beyond the educational sphere. The treatment with students of SSI related to the use of nuclear power, as the one used in this research, besides promoting the development of argumentative skills, can make them more aware of issues related to energy policy, communication of information about nuclear power, environmental policy, social aspects such as energy supply and demand, etc. Regarding the teaching strategy, the roleplaying game used in this study can also be useful to familiarize students with the processes of citizen participation, linked to educational activism [62].

One limitation of this work is that, in the analysis of the PPTs' arguments, a scientific-technological informal reasoning mode has not been established, as can be found in other works, including those referred to in this paper. A review of the analysis carried out from a scientific-technological perspective would be necessary to check whether there are similarities with studies that do analyse this dimension of SSI. Similarly, in future works, it would be advisable to better establish the aspects associated with each informal reasoning mode for analysing the PPTs' arguments, given that authors seem to not follow the same criteria for the configuration of the informal reasoning modes. For example, Crujeiras et al. [5] associate arguments related to human health with social informal reasoning mode, while in the present work such arguments have been classified in the environmental informal reasoning mode. This indicates that some of the factors determining the different results found in the literature stem from methodological aspects derived from the different perspectives applied.

**Author Contributions:** Conceptualization, I.M.C.-L., D.C.-R., M.d.C.A.-E. and M.E.; Data curation, I.M.C.-L. and D.C.-R.; Formal analysis, I.M.C.-L. and M.E.; Investigation, I.M.C.-L., D.C.-R., M.d.C.A.-E. and M.E.; Methodology, I.M.C.-L. and M.E.; Resources, I.M.C.-L., D.C.-R. and M.d.C.A.-E.; Software, I.M.C.-L., D.C.-R. and M.E.; Supervision, D.C.-R., M.d.C.A.-E. and M.E.; Validation, I.M.C.-L., D.C.-R. and M.E.; Visualization, I.M.C.-L. and D.C.-R.; Writing—original draft, I.M.C.-L.; Writing—review & editing, I.M.C.-L., D.C.-R., M.d.C.A.-E. and M.E. All authors have read and agreed to the published version of the manuscript.

**Funding:** This study was supported by the European Social Fund and Spain's National Research Agency through a researcher training contract (PRE2018-083328) as part of the Excellence in R+D project "Developing the competences of secondary and university students in relation to everyday problems through the scientific practices of argumentation, inquiry and modelling" (EDU2017-82197-P), and by the Spanish National Plan R+D+i Project, entitled "Citizens with critical thinking: A challenge for teachers in science education" (PID2019-105765GA-I00).

**Institutional Review Board Statement:** The University of Málaga did not have specific protocols for this type of studies when this research was carried out. Nevertheless, the participants were informed that their data would be treated exclusively for research purposes, and that it would be impossible to identify them in the publications derived from the study.

**Informed Consent Statement:** Informed consent was obtained from all subjects involved in the study.

**Data Availability Statement:** Not applicable.

**Acknowledgments:** The authors of this paper would like to thank Ángel Blanco López, from the University of Málaga, for his help in the development of this study.

**Conflicts of Interest:** The authors declare no conflict of interest.

## Appendix A

Examples of quotes included in each category of the analysis framework. The categories are not mutually exclusive. Text in square brackets has been added for better understanding of the examples.

**Table A1.** Examples of quotes included in each category of the analysis framework.

| Category | Example |
|---|---|
| Environmental | [ ... ] so it is totally good for the environment [ ... ]. (A2, MNP) |
| Climate change | But [nuclear power] generates radioactivity and a lot of things that really affect climate change. (A2, PO) |
| Health | I am totally against nuclear power stations because [they] can cause accidents like the ones we have seen, which are lethal to human health [ ... ]. (A2, PO) |
| Pollution | [ ... ] actually, cars driving around the city pollute more than [nuclear] power stations. (B1, MNP) |
| Resources availability | You cannot have the same number of hours or days of sunshine everywhere [ ... ] (B2, MNC) |
| Resources depletion | [ ... ] no matter how much you develop [the technology], until nuclear power stations work by fusion instead of fission, you are going to have [uranium depletion]. (A2, SE) |
| Safety | Despite what it is said, [nuclear power] is quite safe because there are usually no failures in the reactors. When there are, unfortunately, they are spectacular failures, but there are hardly any failures. (B2, MNC) |
| Waste and waste management | [ ... ] and the waste that is produced needs storage space, [ ... ], and it takes a long time for it to stop being radioactive. (A1, MP) |
| Financial | [ ... ] regarding the economy, [nuclear power] would also be an important aspect for the country. (B2, SE) |
| Continuous production | [ ... ] [nuclear power] works 24 h a day, 7 days a week, and not like solar power, for example [ ... ]. (B2, MNC) |
| Dependence on other countries | [ ... ] on the other hand, we would not have to go to other countries to ask them for oil [ ... ]. (A1, E) |

**Table A1.** *Cont.*

| Category | Example |
|---|---|
| Earn money | You do not build just one wind turbine; you build many wind turbines so that your company . . . has large economic amounts. (A2, MNP) |
| Economic interests/corruption | At no time have I spoken of corruption, only that, if it is stipulated that the duration of nuclear power stations is 25 years, how can there be nuclear power stations that are more than 30 years old, 40 years old, still in operation [ . . . ]? (A1, E) |
| Employment | Renewable energy is also expected to create a lot of jobs. I have a piece of information that in Spain they are going to generate 45,000 jobs. (B2, RS) |
| Price stability | It offers, above all, price stability [ . . . ]. Unlike other traditional fossil energy sources such as oil, where the price varies constantly. (A1, MNP) |
| Production costs | [ . . . ] when you use solar power you really have to make an initial investment to install the solar panels, but once you pay for those solar panels, it does not cost anything else. (B2, SE) |
| Saving money | [ . . . ] this type of [renewable] energy can save up to 200 billion euros per year [ . . . ]. (A2, RS) |
| Viability | [ . . . ] fossil energies can be changed [ . . . ] before 2050 [ . . . ]. (B1, RS Adv.) |
| Social | ——— |
| Awareness/perception on nuclear power | [ . . . ] and the social opposition that this generates, since there is increasing sensitivity on the subject, due to the disadvantages it [nuclear power] has in terms of health or possible accidents [ . . . ]. (A2, PO) |
| Energy supply | For example, a new-generation nuclear reactor fuelled by radioactive waste is capable of meeting the energy needs of the United States for the next 800 years. (A2, NS Adv.) |
| Media | In addition, [renewables] have better press. (A2, PO) |
| Nuclear weapons and terrorism | [ . . . ] [nuclear power] has a kind of relationship with nuclear weapons that would be interesting to consider. (A2, PO) |
| Other uses | [ . . . ] I do not know if you know that without nuclear energy it would not be possible to carry out cures such as chemotherapy processes or the famous X-rays and radiopharmaceuticals that are necessary for many people to live. (B1, NS) |
| Politics | Another aspect in our favour is that the European Union proposes subsidies to countries that use solar power. (B2, SE) |
| Reduce consumption | I believe that what needs to change in our society is consumer demand. (B1, E) |
| Resettle | Furthermore, [nuclear power] [ . . . ] would be a good way of populating the areas of the interior of Spain that are depopulated. (B2, MNC) |
| Technological development | Spain is currently one of the pioneers in Europe in specialised nuclear power technology. We are now an exporter of this technology, which moves many sectors [ . . . ]. (A1, MNP) |
| Unknown aspect | [ . . . ] but it also has many disadvantages that you have to carry over time [it is not possible to know whether he is referring to environmental, financial or social disadvantages]. (A2, SE) |

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
