# Peer review of "Analysis of the Informal Reasoning Modes of Preservice Primary Teachers When Arguing about a Socio-Scientific Issue on Nuclear Power during a Role Play"

_sustainability, doi:10.3390/su15054291_

Round 1

Reviewer 1 Report

In this study, the main question addressed by the research is:

Nuclear power use is a socio-scientific topic that sparks debate in a variety of sectors, including health, the environment, jobs, and energy supply, and it is the subject of both arguments for and against. In order to better understand preservice primary teachers' informal reasoning styles, as well as the potential consequences of the roles they defended and the role play's staging, this paper looks at the specific issues raised by these teachers as part of their arguments during their participation in a role play on the SSI of nuclear power stations closure in Spain. Four role plays were examined for their transcripts in order to achieve this.

I found the paper interesting to read and enjoyed it. The topic is relevant to the field. 

I found the study adequate.

The conclusion is well written.

I found that the references are appropriate.

I have no additional comments on the tables and figures.

Reviewer 2 Report

A clear overall paper. There were occasional grammatical errors that should be addressed, but nothing major. In this instance, I had no further comments to make on the manuscript. Beyond some minor grammatical errors the paper was well written. It provides a clear rationale for examining role play skills in the context of SSI through the particular use of nuclear power. The methods were clearly described, with good detail in the analysis procedures. The results were clearly articulated, and sensible recommendations for practice in pre-service teacher education (and, thereby, potentially for classroom practice) were made by the authors.

Reviewer 3 Report

The paper presents an interesting topic and relevant to the current issue on energy security and climate change mitigation. It perfectly fits with the aims and scope of the Special Issue of Sustainability: "Sustainability and Citizenship: Integration of Socio-Scientific Issues in Science Education". On the other hand, the writing should be improved to make the ideas comprehensible to interdisciplinary readers of the journal. Specific comments are as follows:

1. All acronyms should be defined on their first use.

2. Start the Introduction with a general research background. Followed by the problem/motivation why the study should be conducted. Then, the Literature Review, gap and proposed contribution as presented in 1.1 to 1.3. If these are too long, 1.1 to 1.3 can be written in another section dedicated to Review of Related Literature.

3. In the Methodology, Ethical Considerations should be discussed since the study involved students as subjects. If possible, the parent's consent is needed if the participants are minors.

4. Do not cut words in the tables.

5. Figures should be improved in terms of the quality and color. Avoid using abbreviations, otherwise, define them in the notes (below the title of the figure). Same with tables

6. To increase the significance of the study, the discussion should include implications beyond education such as energy policy, education and information communication about nuclear, environmental policy, social aspect, etc.

7. References must be improved by citing the most recent (5 years or newer) and relevant literatures.

Round 2

Reviewer 3 Report

The authors carefully addressed all the comments and suggestions of the reviewers and made significant changes to improve the manuscript.

I have no further comments and am looking forward to read the published version of the manuscript.